# Cowboys: Abstract Expressionism, Hollywood Westerns, and American Progress

**Justin Kedl**

Independent Researcher, Beverly, MA 01915, USA; justindkedl@gmail.com

**Abstract:** Abstract Expressionism has been influenced heavily by the popular theory of America's undying, progressive spirit, originally conceived by Frederick Jackson Turner and given its most potent form in Western films. Turner's "Frontier Thesis" was embodied in stories of John Wayne and other cowboy heroes taming the supposed edges of civilization. The mythic West as constructed by Turner and these films cemented American identity as one of exploration and innovation, with the notable condition of Indigenous Americans ceding their sovereignty. While Abstract Expressionism was commonly connected to the mythic West through the origin stories of Jackson Pollock and Clyfford Still, the critical understanding of this movement as the height of painterly achievement built on Native American precedents evinces a deeper connection to Turner's popular Frontier theory. As critics like Clement Greenberg cast flatness as the last frontier of painting, and as artists like Pollock and Barnett Newman claimed Native American ritual practices as a part of their aesthetic lineage, Abstract Expressionism proved as effective as Hollywood Westerns in corroborating and perpetuating the idea of America's frontier spirit.

**Keywords:** Abstract Expressionism; Western films; Frederick Jackson Turner; American history; historiography; visual culture





## 1. Introduction

We have in our minds a certain image of the Bohemian artist: lonesome by choice, pensive, but passionate in the throes of inspiration from the blue. A notable example of this comes from Ed Harris' 2000 film *Pollock*. Harris, playing a brooding Jackson Pollock tortured by his career, the art world, and his alcoholism alike, agonizes over his mural for Peggy Guggenheim in one of the film's critical moments. We see him crouched motionless in his studio for days, staring at a blank canvas, unable to begin. Suddenly and with a rushing score, he attacks the canvas, and in a montage of flurried activity Jackson Pollock's monumental *Mural* (1943) is completed (Harris 2001).

Indeed, the careers of the Abstract Expressionists did much to cement this trope even as films like *Pollock* perpetuate it. The oft-repeated photograph of the Abstract Expressionist painter standing arms folded before his work, defying the camera and any who dare to question it, went hand-in-hand with the lofty intellectualism of artists like Clyfford Still and Barnett Newman. Peggy Guggenheim herself perpetuated the mythic account of Pollock creating *Mural* in the span of a few hours, when in reality it took him months to complete (Rushing 1995, p. 186; Fontanella 1943; Rubin 1967).

While largely untrue, this trope, admittedly, makes for spectacular filmmaking. It is notably similar to a trope of another genre. The cowboy hero of classic westerns, *à la* John Wayne, is similarly lonesome, brooding, at times passionate. The heroic cowboy and the heroic artist alike contend with cosmic forces of nature or of inspiration in a self-imposed holy solitude before the expanse of the canvas or the American West (Figure 1).

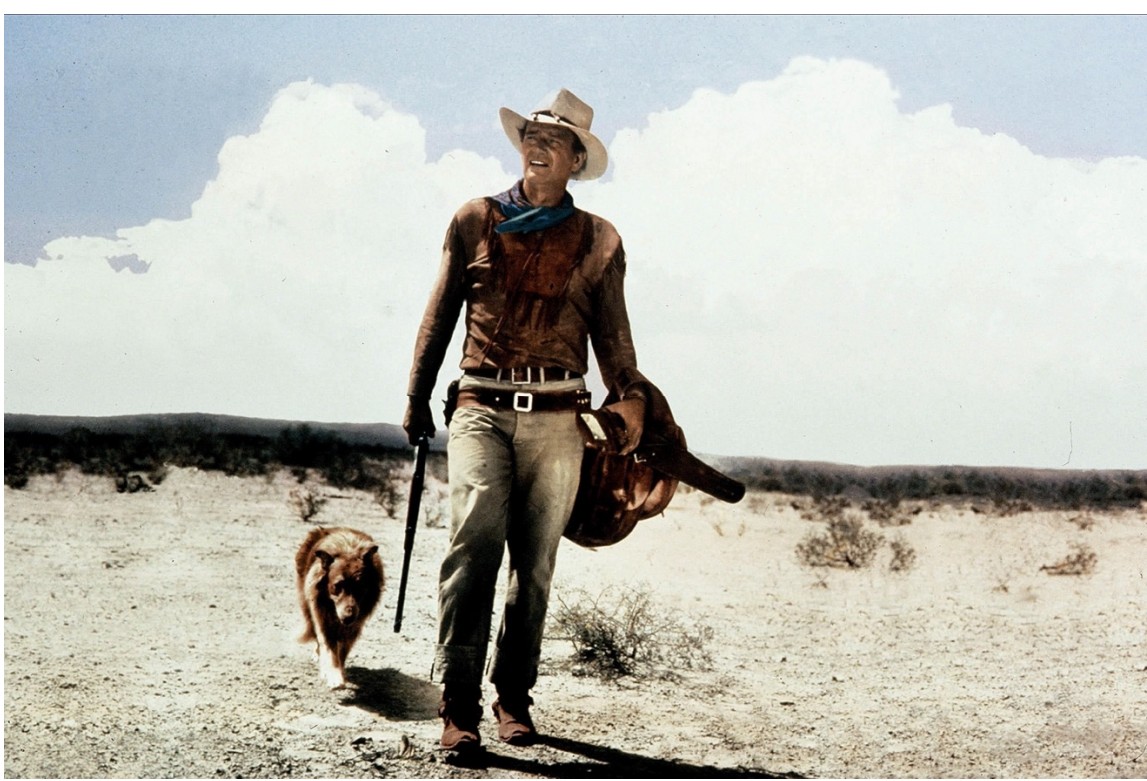

**Figure 1.** Promotional image of John Wayne in *Hondo*, directed by John Farrow. 1953; Hollywood, CA: Paramount Pictures, copyright 2023. Film still © Allstar Picture Library Limited/Alamy Stock Photo. All images used by permission.

Several scholars have already identified an affinity between certain Abstract Expressionist paintings and what we know as the American West. At the core of such affinities, however, lies an idea about American progress that has been put forward in both the realms of American art and entertainment. I propose that our ideas about Abstract Expressionism have been influenced heavily by the theory American progress, given its most potent form in Western films, as it was originally conceived by Frederick Jackson Turner in what has become known as his "Frontier Thesis." The mythic West as constructed by these films and other popular media—an epic landscape of cowboy heroes taming the supposed edges of civilization—cemented American identity as one of exploration and innovation, with the notable condition of Indigenous Americans ceding their sovereignty. As critics like Clement Greenberg cast flatness as the last frontier of painting, and as artists like Pollock and Barnett Newman claimed Native American ritual practices as a part of their aesthetic lineage, Abstract Expressionism both corroborated and perpetuated the popular fantasy of America's frontier spirit.

While I will be looking at a few particular paintings by the Abstract Expressionists, I am more interested in Abstract Expressionism as a multivalent cultural phenomenon: the paintings themselves, as well as the stories told about the artists and the common understanding of their work. Abstract Expressionism as we understand it has been constructed as much by artists as by institutions, critics, the government, and the public alike. Just so for the idea of the American West: created as much by historical figures as the scholars, artists, moviemakers, Americans, and non-Americans who continue to tell stories about it. As these two cultural miasmas takes shape, they attain degrees of similarity which complicate any linear narrative of one person directly influencing another, one idea acting on another. Such striking parallels speak instead to broader cultural notions which have directed actors in both circles.

## 2. The New American Painting

In 1958, Alfred Barr brought a traveling exhibition of Abstract Expressionism titled "The New American Painting" to cities around Europe. Many of the critics equated these paintings with the unique circumstances of the country they came from. The sizes of the canvases were frequently compared to the size of the United States: "This is a demonstration of strength in proportion to the size of the United States. The Biggest In the World [sic.] . . . " wrote one critic in Brussels; " . . . their enormous dimensions make us sense what goes on within an American painter, faced with the immensity of his continent and his growing history" wrote another critic in Berlin. Recalling Buffalo Bill Cody's raucous performances that similarly traveled Europe, another critic in Brussels disparagingly compared the touring exhibition to "a wild-west show." (Landau 2005, pp. 221–24).

These critics were not out of place in making such comparisons as American critics themselves were writing articles about the "New *American* Painting", "*American*-Type Painting," "*American*-Action Painting."[1] As the national economy and art world alike ascended the geopolitical ranks after years of European preeminence, artists and critics were eager to say that their work was unique, bred of American soil. The US Government itself took an interest: by way of the Rockefellers and the International Program of the Museum of Modern Art, the CIA and the USIA sponsored several touring exhibitions of Abstract Expressionist art to showcase the artistic freedom available in a democratic society to European nations in the wake of Naziism and the shadow of the Soviet Union. Barr's "The New American Painting" was one such exhibition (Cockroft 1985, pp. 130–31; Kozloff 1973).

This hearty American read is particularly relevant when it comes to Jackson Pollock, whose work was included in the exhibition. Given Pollock's rapid rise to stardom amidst the New American Painters, his origins in and affinity for the American West were told and retold. Pollock was born in Cody, Wyoming, and by all accounts proud of it. An early Regionalist painting by Pollock, *Going West* (c.1934–35), shows a dark yet romantic scene of a man with a broad-rimmed hat driving a team of horses and two wagons into the hills. Pollock's scene is nostalgic, a hazy dream of hard work and bravery before dark nature (Figure 2). Such a painting makes sense not only in the context of Pollock's childhood on dairy and produce farms in the early twentieth century, but also given his early education with the American Regionalist Thomas Hart Benton. Benton's own paintings of down-home, pioneer America were immensely popular in the 1930s. A young Pollock saw in them not merely the depiction of his home culture, but also a concern for the working-class of Western America amongst whom he was raised. Pollock would move to New York in 1930 to become a student and close family friend of Benton's, finding a kinship in their love for Western America (Doss 1991, pp. 311–29). Even though Pollock would later go on to say that his Regionalist education under Benton's tutelage was "important as something against which to react very strongly," his later abstractions still carry what Erika Doss calls a "Bentonesque form . . . in color, size and energy . . . ." (Landau 2005, p. 132; Doss 1991, p. 348). His early Western, Regionalist influences, then, directed the form of his later abstractions.

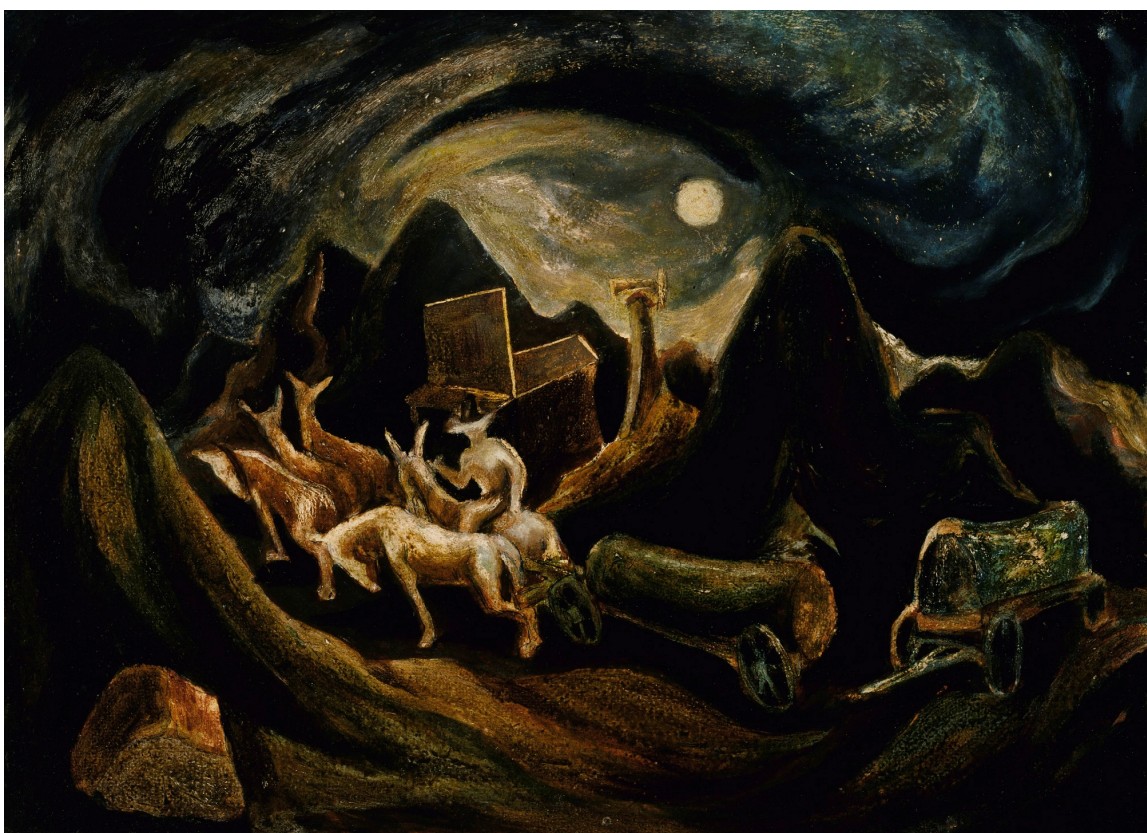

**Figure 2.** Jackson Pollock. *Going West*, c.1934–38, oil on gesso on composition board, 15 1/8 × 20 3/4 in. Smithsonian American Art Museum, Gift of Thomas Hart Benton. Washington, DC. © 2023 The Pollock-Krasner Foundation/Artists Rights Society (ARS), New York. https://americanart.si.edu/artwork/going-west-19820, accessed on 3 October 2022.

Pollock's cowboy affinity was well known among the other artists and critics of the New York school—although some, like Harold Rosenberg, were more skeptical of Pollock "playing cowboy" than authentically being one (Desmaris and Smith 2017, p. 136). Nonetheless, Pollock's cowboy reputation became well-enough known that the magazine *Arts & Architecture* made it a point to ask him about his Western origins in a questionnaire for the 1944 issue. " . . . I have a definite feeling for the West:" Pollock answered, "the vast horizontality of the land, for instance; here [in New York] only the Atlantic Ocean gives you that." (Landau 2005, p. 132). As Pollock himself connects his sprawling compositions with the "wide open spaces" of the American West, his European audience was not wrong to draw the same conclusions. William Rubin, however, later argued that the European fanaticism for cowboy Pollock was blown far out of proportion by "the cult of Hollywood Westerns celebrated by the young critics of the *Cahiers du cinema*" magazine (Rubin 1967).

As with any person who attains a degree of celebrity, there is a mythic quality in the construction of an image apart from the artist himself. Pollock's cowboy image may not have always been accurate, but it was created by critics at home and abroad as well as by him, and based in a certain amount of fact. He was born in Wyoming and raised between there, Arizona, and California for his entire childhood (Desmaris and Smith 2017, p. 136). An early photograph of a teenage Pollock shows him in stereotypical cowboy garb, complete with boots and a broad-rimmed hat, kneeling and aiming a rifle out of frame (Figure 4).[2] Compare this to any costume John Wayne has worn, and the myth of the "cowboy painter" is at least understandable (Figure 3) (Rubin 1967).

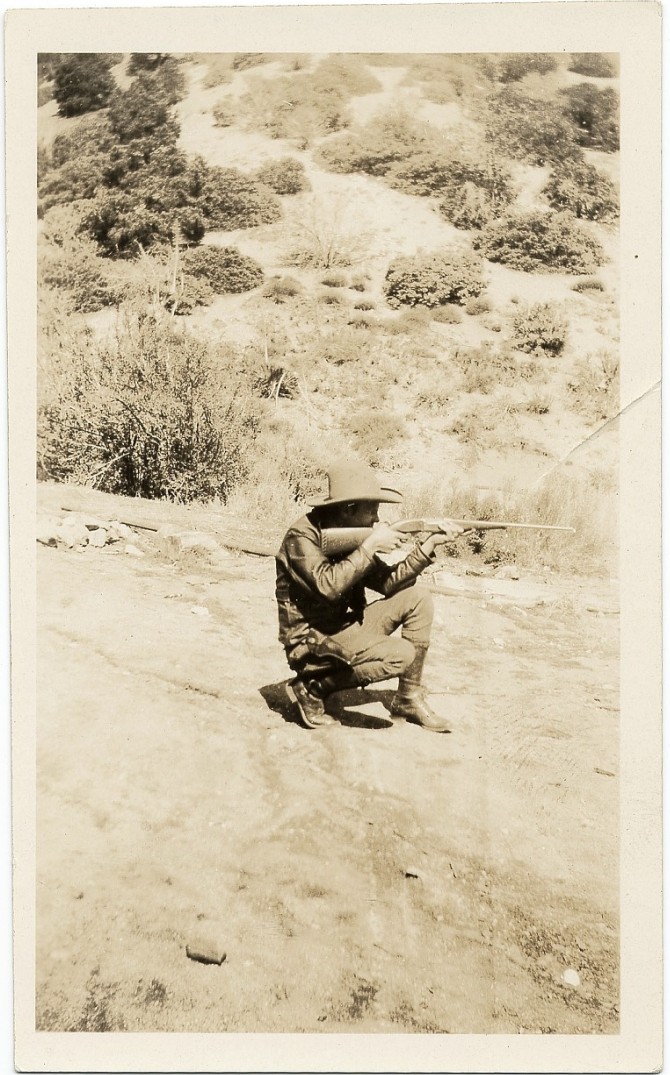

**Figure 3.** Jackson Pollock hunting, ca. 1927. Photographer unknown. Jackson Pollock and Lee Krasner papers, circa 1914–1984. Archives of American Art, Smithsonian Institution. https://www.si.edu/object/jackson-pollock-hunting%3AAAADCD_item_3862, accessed on 3 October 2022.

Another photo of a teenaged Abstract Expressionist is nearly symmetrical to the photograph of the young Pollock. Sometime between 1912 and 1920, Clyfford Still posed for a photo in his scout uniform, also in boots and a broad-rimmed hat, also kneeling and pointing a rifle out of frame (Figure 4). The resemblance to Pollock's photograph is uncanny, and for good reason. Still was also born and raised in various cities throughout Western America and Canada: North Dakota, Washington State, Alberta, and California. His experience of the cultural and geographic West also deeply affected his work: we see the same sense of scale, with a unique interplay between horizontal and vertical forms that mirrors the visual experience of the great plains (Anfam 2015, pp. 580–82). The grain elevators rising from the fields of Still's early landscapes evolve by degrees into the craggy masses of his mature period. These canvases are marked by vertical brushstrokes and broad expanses of color, like buildings or rock formations rising from the prairie. There is a notable similarity, too, between Still's mature work and the stark geography of places like Monument Valley, Utah, made famous as "the archetypal West" by the director John Ford who used the location for seven different films (Figure 5) (Perrottet 2010).

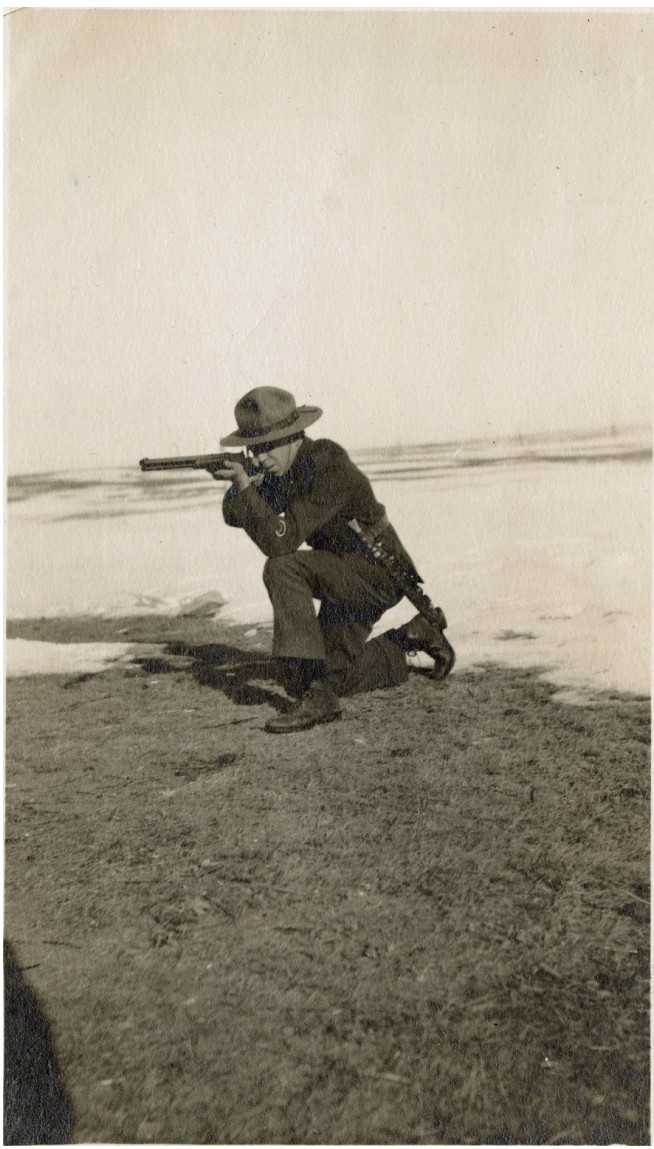

**Figure 4.** Early portrait of Clyfford Still with scout's uniform and rifle, 1912–1920. Photographer unknown. © 2023 City & County of Denver, Courtesy Clyfford Still Museum. Museum ID CPSA.F001.S003.SB001.B001.040. Accessed on 3 October 2022.

Scholars today still cite both Pollock's and Still's cowboy origin stories, however mythic they may have become. David Anfam, formerly the Senior Consulting Curator for the Clyfford Still Museum, writes that the two painters shared a "romance of the American West" that defined their friendship and their careers (Anfam 2015, p. 12):

> Consider Pollock's oft-quoted remark: 'I have a definite feeling for the West: the vast horizontality of the land, for instance; here only the Atlantic Ocean gives you that.' Now compare it with one of several comments by Still: 'It was a grand feeling Pollock and I felt out in the West, looking over the mountains, being able to move any number of miles. Where I come from you don't die for the status symbol or the fancy Fifth Avenue parade.' The *topos* at stake is as old as Europe's first brushes with the Americas, as corny as a song such as Cole Porter and Robert Fletcher's 'Don't Fence Me In' (1934) and as seminal to American culture as a book such as Charles Olson's *Call Me Ishmael: A Study of Melville* (1947) (Anfam 2015, p. 18).

Although Pollock's and Still's paintings are vastly different, each of these artists allowed the scale of the West to manifest through their large canvases. The plain-speaking Western attitude mentioned here by Still is also evident in both: paint acting as paint rather than creating any kind of illusory scene. It is significant that these two painters—each of whom was championed by critics as influential, if not superlative—were both so impacted by the American West. *LIFE* magazine famously questioned if Pollock was "the greatest living painter in the United States," while Clement Greenberg, one of the pillars of Abstract Expressionist theory, called Still "one of the most important and original painters of our time—perhaps the most original of all painters under fifty-five, if not the best." (LIFE Magazine 1949, p. 42; Greenberg 1993, p. 208). As an educator, Still also had a very direct hand in shaping Abstract Expressionism's second generation. Notable, too, is the fact that Pollock and Still were among the few artists mentioned by name by those European critics writing on "The New American Painting" exhibition. If the American West shaped these two vital figures of Abstract Expressionism—either in fact or perception or both—then we can say that the American West to a degree shaped the movement as a whole.

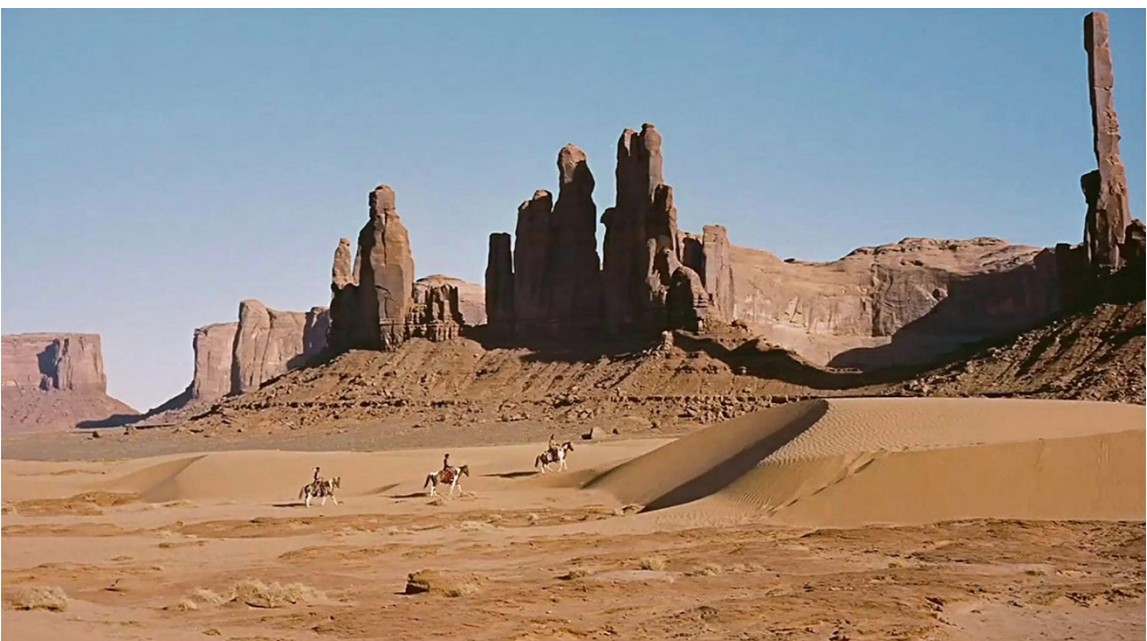

**Figure 5.** Film still from *The Searchers*, directed by John Ford. 1956; Burbank, CA: Warner Bros. Pictures, copyright 2023. Film still © Pictorial Press Ltd./Alamy Stock Photo.

"The West," however, is an ambiguous term. There are specificities to Pollock's and Still's lives—their birthplaces, their formative experiences in prairies and deserts—but time and again critics generalize to speak of their origins in "the West." As used by scholars like Anfam and by the artists themselves, it references both a broad geography as well as a subset of culture, an aesthetic as much as a way of life. Much like the origins of Pollock and Still, "the West" as we understand it has been shaped equally by historical fact and legend, particularly legends as they appear on the silver screen.

The reader will note the similarities already drawn to images from Western films: John Wayne in *Hondo* (Farrow 1953), Monument Valley as captured by director John Ford. There is a consistency in Wayne's appearances from earlier films like *Stagecoach* (Ford 1939) to later films like *Hondo* and *The Searchers* (Ford 1956): the hat, the boots, the gun. Ford's continuous use of the unmistakable Monument Valley to represent many different Western territories—some actual and some fictional—speaks to the construction of a Western nowhere: a mythic landscape that corresponds not to fact, but to an *idea* of what "the West" was. The literary critic Scott Eyman says that Ford "gave the Western a look and feel and, in the process, did the same for the country he loved and served so passionately" (Desmaris and Smith 2017,

p. 119). The sheer popularity of Ford's and Wayne's work created a consistent Western hero in a consistent Western landscape.

Pollock and Still's "Western-ness" resembles this type created by Hollywood as it does for many Americans and non-Americans, and has done for over a century. Since at least the late 1800s, America has benefitted from sensationalizing and exporting the drama of settler and homesteader life to audiences on the East coast and beyond. Buffalo Bill Cody, mentioned previously, was perhaps the most famous of these traveling cowboy-showmen, dramatizing his experiences as a military scout on the great plains. His show eventually came to feature the real scalp of a Cheyenne man named Yellow Hair, who Cody purportedly killed to avenge Custer's defeat at the Battle of Little Bighorn. As Dr. Josh Garrett-Davis, an Associate Curator at the Autry Museum of the American West argues, the blending of Cody's military career with his onstage personality—going so far as to kill a man to gain a new prop—is emblematic of how these shows and other Western media have made "myth and reality . . . really hard to disentangle" when it comes to the American West (Jones and Rau 2021, 42:00–44:09). These myths carried over from Buffalo Bill to John Ford, and continue to build an imaginary West that looks so similar to Pollock's *Going West* painting.

There is, too, an existential wrestling common to both. Harold Rosenberg, one of the other seminal theorists of Abstract Expressionism, argued that each painting encapsulated the artist's action: the final canvas standing for the performance of the artist in creating the work. "What was to go on the canvas was not a picture but an event," he famously wrote. "The painter no longer approached his easel with an image in his mind; he went up to it with material in his hand to do something to that other piece of material in front of him. The image would be the result of this encounter." (Landau 2005, p. 190). With this emphasis on the artist's process rather than the image they created, Rosenberg argued that "the act-painting is of the same metaphysical substance as the artist's existence" (Landau 2005, p. 191). The marks made on the canvas would inevitably be influenced by the artist's psychic state; therefore, each painting represented the artist's exploration of their own "private myths" (Landau 2005, p. 193). Pollock in particular embraced this interpretation of the New American Painting, given his oft-cited interest in Jungian theory and his training with the Surrealists who fled to New York during the Second World War (Rubin 1967). Mary-Dailey Desmaris points to this commonality between Pollock and the mythic West as the cowboy hero often confronts something of himself in his struggles against the wilderness: " . . . in Westerns of the classic period (generally considered to be from 1946 to 1960), the relationship between the cowboy and the landscape came to thematize the subject's struggle to wrest control from the chaos within." (Desmaris and Smith 2017, p. 141). Wayne's character in *The Searchers*, for example, confronts his own demons but fails to fully overcome them. Mired in his own prejudice and unable to rejoice in the rescue of his niece at the end of the film, Wayne turns back into the desert in a dramatic final shot as the rest of the family goes into the safety of the house.

In this scene as in many other iconic shots from classic Westerns, the sheer scale of the landscape accentuates the psychic drama as Wayne's figure diminishes against the backdrop of a distant mountain range. Given the grandeur of the filming locations, the Western genre proved to be effective in testing early widescreen technologies: Fox Film Corporation's *The Big Trail* (1930) and MGM's *Billy the Kid* (1930) each used early versions of widescreen shot on 70 mm film, twice the size of standard formats at the time. The larger format, directors argued, provided a "'panoramic expanse' appropriate to the material" (Belton 1992, pp. 48–50). While these early formats faltered before widescreen technology could catch up to the studios' visions and the public's tastes, directors in the 1950s used improved widescreen film to capture the sweeping grandeur of the American landscape in Western films as well as in documentaries and travelogues about the American West (Belton 1992, pp. 89–94). Just as the "vast horizontality of the land" impressed itself upon Pollock, Hollywood directors were similarly affected to communicate the majesty of the Western landscape with wider film stock. Both recall the tactics of Romantic-era painters to

achieve the *sublime*: awe before magnificent, cosmic forces. It is doubtful that any of these Hollywood directors claimed the sublime as either the goal or outcome of their films. Even so, Robert Rosenblum's assessment of the sublime scale of Abstract Expressionist painting could just as easily be applied to the mountain-magnifying tactics of Western film: " ... a breathtaking confrontation with boundlessness in which we also experience an equally powerful totality ... " (Rosenblum 1961). As large scale and existential drama were core traits for painters beyond Pollock and Still, the similarly grand scale and dramas of the Western rhyme with the New American Painting as a whole.

This does not necessarily mean that all the Abstract Expressionists were big Western fans, although Pollock at least was said to be "an avid watcher of Westerns" (Desmaris and Smith 2017, p. 136). These painters likely had some contact with the genre as 30 percent of all motion pictures produced by major studios between 1947 to 1950—a peak time for Abstract Expressionism—were Westerns (Corkin 2004, p. 2). Rather, these similarities speak more to engrained ideas about American progress that influence both fine art and popular culture.

### 3. Frontiers

In 1893, a professor of history at the University of Wisconsin named Frederick Jackson Turner read his paper "The Significance of the Frontier in American History" before an audience at the World's Columbian Exposition in Chicago. Turner theorized that America was developed through a drive to tame the edges of civilization in an ever-advancing march Westwards until there was no more wilderness to tame. By tracking how America continually civilized "successive frontiers," Turner argued that scholars could observe all the major stages of societal evolution (Faragher and Turner 1994, p. 36). Turner writes:

> The United States lies like a huge page in the history of society. Line by line as we read this continental page from West to East we find the record of social evolution. It begins with the Indian and the hunter, it goes on to tell of the disintegration of savagery by the entrance of the trader, the pathfinder of civilization; we read the annals of the pastoral stage in ranch life; the exploitations of the soil by the raising of unrotated crops of corn and wheat in sparsely settled farming communities; the intensive culture of the denser farm settlement; and finally the manufacturing organization with city and factory system (Faragher and Turner 1994, p. 38).

According to Turner, this frontier spirit began with the first European settlers meeting the demands of the "wilderness" present on an "inert continent" (Faragher and Turner 1994, pp. 33, 41). Once this rough-and-tumble character was developed, it recapitulated itself by pushing Euro-Americans ever into the Western wilderness. The frontier, then, acted as a "consolidating agent," thus unifying the large territory we now know as America rather than establishing many smaller, independent territories (Faragher and Turner 1994, p. 42).

Turner believed that the American spirit had been so defined by the experience of the Western frontier that this country could not help but find new wildernesses to explore even after settler and homesteader life had ended. "He would be a rash prophet who should assert that the expansive character of American life has now entirely ceased," Turner wrote towards the end of his paper. "Movement has been its dominant fact, and, unless this training has no effect upon a people, American energy will continually demand a wider field for its exercise" (Faragher and Turner 1994, p. 59). This notion of a "wider field" has encouraged everything from American imperialism to the space program. Inspired by Turner, President Woodrow Wilson, among others, argued that America had to seek "new frontiers in the Indies and in the Far Pacific" (Faragher and Turner 1994, p. 80). In announcing plans for the moon landing during a speech to Rice University on 12 September 1962, President John F. Kennedy said,

> Surely the opening vistas of space promise high costs and hardships, as well as high reward. So it is not surprising that some would have us stay where we are a little longer to rest, to wait. But this city of Houston, this State of Texas, this country of the United States was not built by those who waited and rested and

wished to look behind them. This country was conquered by those who moved forward—and so will space (NASA Video 1962).

Not only was Turner's "wider field" an inevitable outcome of the frontier experience, he saw it as necessary for his country's survival. As the geographic frontier closed, Turner posited that America must be "thrown back upon itself" to innovate within its current conditions in order to keep this unifying spirit alive (Faragher and Turner 1994, p. 8).

There are of course many problems with Turner's theory, some of which were recognized in his own day. Upon the first reading of his paper, Turner insisted that the frontier had officially closed, although new settlements were still founded in the geographic west in the following years. Turner's definitions of a "frontier line" as well as "unsettled territory" are so ambiguously defined that several American territories could have been classified as "unsettled" well into the twentieth century (Faragher and Turner 1994, p. 6). Scarcely mentioned in Turner's account are other settling groups who moved up from Mexico or Eastwards from the West Coast (Jones and Rau 2021, 21:06–22:03). Most glaringly, and as will be discussed later in further detail, the American continent was not *inert* or *uncivilized* as Turner claimed, but was of course home to Indigenous nations who had been living on and cultivating the land for millennia.

Despite these many issues and the lukewarm reception Turner's paper originally received at the exposition, his "Frontier Thesis" quickly became the standard narrative of American development. Scholars and politicians alike accepted Turner's views, including soon-to-be-U.S. president Theodore Roosevelt and later Presidents Wilson and Kennedy, as discussed above. It was accepted by educators and taught in classrooms around the country (Faragher and Turner 1994, pp. 1–2, 8). Scholar John Mack Faragher, among others, argues that "Turner's essay is the single most influential piece of writing in the history of American history." Turner's thesis, Faragher goes on to say, "has been played out in a fascinating interchange of ideas and images among movies, popular history, and political rhetoric" (Faragher and Turner 1994, pp. 1, 230).

Faragher is one among many scholars to acknowledge Western films and other popular media as major vehicles of the Frontier Thesis.[3] These films do not consciously set out to indoctrinate the masses with Turner's ideas; however, as the genre necessarily deals with the gradual settlement and development of America, it cannot help but align with a narrative which for nearly a century shaped our understanding of this period. In most classic Westerns, the cowboy hero and his compatriots overcome the challenges of the American wilderness—often equated with "savage" indigenous people—to establish the beginnings of white society. In *Stagecoach*, Wayne's character Ringo the Kid saves both the money and most of the American citizens aboard a stagecoach from an aggressive band of Apaches. In *Broken Arrow* (Daves 1950), Tom Jeffords, played by Jimmy Stewart, establishes a treaty with the Apaches to allow for mail and wagons to make their way safely from town to town. Even in the heroic tragedy of *Fort Apache*, as Col. Thursday played by Henry Fonda dies in a thinly veiled reference to Custer's defeat at Little Bighorn, his perceived heroism unifies the cavalry and gives them a reason to declare war against the Apaches (Ford 1948). Mail, money, and the US Cavalry are all important institutions of white society, "civilizing" forces that are brought to bear by the cowboy hero on the land and the native inhabitants alike before he vanishes back into the sunset.[4]

Speaking of the cliché'd sunset-silhouette, this American *rückenfigur* is a striking visual manifestation of the Frontier thesis. The original *rückenfigur* of Romantic-era landscapes dwindles in comparison to their immense, awe-inspiring surroundings. Here, the effect is the same: the European Alps and the Baltic Sea instead replaced by America's wide open spaces. Whether the hero is on foot or on horseback, alone or with the leading lady, at the head of a wagon train or the cavalry, most films end with the hero returning to the wilderness.[5] Just as Turner argued, the cowboy hero and the American people both will continually push into the West for the sake of American progress.

Given the conscious nationalization of the New *American* Painting, it is no surprise that this same frontier mindset found its way into Abstract Expressionist theory. Clement

Greenberg, already mentioned, was a dogmatic theorist whose historicism popularized a view of Abstract Expressionism as painting's furthest evolution. Greenberg argued that every art form under modernism was seeking to "determine, through its own operations and works, the effects exclusive to itself" to assert its relevance in the modern world (Greenberg 1993, p. 86). The importance of the following oft-quoted passage to the critical understanding of Abstract Expressionism cannot be overstated:

> It was the stressing of the ineluctable flatness of the surface that remained, however, more fundamental than anything else to the processes by which pictorial art criticized and defined itself under Modernism. For flatness alone was unique and exclusive to pictorial art. The enclosing shape of the picture was a limiting condition, or norm, that was shared with the art of the theater; colors as a norm and a means shared not only with the theater, but also with sculpture. Because flatness was the only condition painting shared with no other art, Modernist painting oriented itself to flatness as it did to nothing else (Greenberg 1993, p. 87).

The American painting of the 40's and 50's, Greenberg argued, looked the way it did because it had to find its own final frontier, the domain which it and only it could suitably capture and critique. Painting in the global West, which had defined itself by the representation of illusory space on a flat surface for hundreds of years, had gradually been dissolving its own conventions through the early stages of modernism. Greenberg argued that the Abstract Expressionists were the first to acknowledge the most basic tenet of painting: putting marks on a flat surface. Their all-over compositions stressed the surface itself rather than any illusory space. Such a progressivist view of art history comes from an evolutionary mindset cultivated by European movements, their proliferation of manifestoes proclaiming why their work had evolved beyond their predecessors. Greenberg became unique in declaring that he and his countrymen had found the *end* of painting: pushing into the final frontier of flatness as Turner's pioneers had pushed into the furthest reaches of the American West.

The core overlap between Abstract Expressionism and the mythic West, then, is this idea of American "frontiering." It forms the backbone of the mythic American West as understood by Hollywood and by Pollock and Still, as well as Greenberg's decisive championing of Abstract Expressionism as painting's furthest evolution. There is also a common consequence of the Frontier mindset in these circles. Given Turner's equivocation of Native American society with the "wilderness," calling Indigenous nations "inert" or "savage," the United States' fraught relationship with Native America extends as much into Abstract Expressionism as it does into Hollywood Westerns.

### 4. The "Indian Question"

There is a long-standing discourse in American art which primitivizes Native American spiritual and creative practices to serve modern American culture. In attempting to differentiate themselves from European Modernism, several American artists in the early twentieth century looked to Native American practices as their new cultural patrimony. Part of the migration which brought Georgia O'Keeffe to Santa Fe centered around an "Indianism" wherein artists and others sought "life-changing experiences through their contact with [Native] dances and ceremonies at the pueblos." (Corn 1999, p. 254). With his painting *Indian Fantasy* (1914), Marsden Hartley claimed himself an aesthetic heir of Indigenous American art: what he called "the one American genius we possess" (Corn 1999, p. 255). Romanticist, primitivist language of the "noble savage" made Indigenous Americans simultaneously holier and lesser, sought to pass on their spiritual knowledge before vanishing into the annals of history (Corn 1999, p. 273).

It was not entirely unprecedented, then, for many early Abstract Expressionist painters to take interest in the 1941 MoMA exhibition "Indian Art of the United States." The popularity of this exhibition among artists and the general public attests to a "wider American interest in Indian culture" at the time (Rushing 1995, p. 173). In particular, three major figures of the movement—the cowboys Pollock and Still, as well as the erudite

Barnett Newman—expressed critical interest in Indigenous art forms which impacted both their work and its reception.

To begin with Pollock: much was made and continues to be made of his interest in Native American art. In his aforementioned questionnaire for *Arts & Architecture*, when asked about the influence of the American West on his work, Pollock instantly brought up his inspiration from Native American sources:

> I have always been very impressed with the plastic qualities of American Indian art. The Indians have the true painter's approach in their capacity to get hold of appropriate images, and in their understanding of what constitutes painterly subject-matter. Their color is essentially Western, their vision has the basic universality of all real art (Landau 2005, p. 132).

As Pollock himself attested, his technique of painting on the floor with a stick was inspired by Navajo sand painting: "On the floor I am more at ease. I feel nearer, more a part of the painting, since this way I can walk around it, work from the four sides and literally be *in* the painting. This is akin to the method of the Indian sand painters of the West" (Chipp 1968, p. 546).[6] While most scholars cannot fail to mention Pollock's relationship to Native American art, W. Jackson Rushing gives these influences a comprehensive assessment in his book *Native American Art and the New York Avant-Garde* (Rushing 1995). As Rushing demonstrates, Pollock's interest in Native American mythology combined with his study of Jungian theories to give the artist a library of totemic forms for use in the automatic processes of his early work. *Guardians of the Secret* (1943), for example, is not only covered in a petroglyph-like scrawl, the composition also resembles an altar from the Pueblo people, flanked by two figures resembling totem poles from the Indigenous cultures of the Northwest coast (Figure 6).[7] Sections of the black lines undergirding the Guggenheim *Mural* also resemble traditional depictions of the Kokopelli spirit from prehistoric cultures in the geographic Southwest (Rushing 1995, pp. 177–78). The line work of these symbols and totems were progressively loosened into the drizzles and drips of his mature period. Insisting that these interests originated in firsthand experience in the American West, Pollock was not shy about his affinities for the arts of Native cultures across America (Landau 2005, pp. 132–33; Rushing 1995, pp. 169–70).

Clyfford Still was also impacted by a personal experience with Native American culture. From 1936 to 1939, Still took several trips to the Colville Indian Reservation to draw portraits and scenes of Native American life. The experience was so impactful for him that he petitioned Washington State College, at which he was a junior faculty member, to establish an art colony there (Miller 2015). While documenting the lives of the people, Still also watched and recorded the construction of the Grand Coulee Dam on the reservation. By eliminating the salmon runs on which they depended, the dam had tragic, far-reaching effects for the Colville people (Clyfford Still Museum 2016, 2:23–3:49). The devastation struck a cord with Still, whose own young adulthood on an isolated Canadian homestead was marked by years of drought, combining with the Great Depression to create an economic and ecological disaster (Landau 2005, p. 581). The tragedy of these two experiences are married in the elongated, expressionistic figures of his early work, which eventually morphed into his mature abstractions. While Still did not loudly proclaim the influence of Indigenous cultures on his paintings as Pollock did, his experience with Native people at the Colville reservation is an important moment that directed the development of his later, celebrated oeuvre.

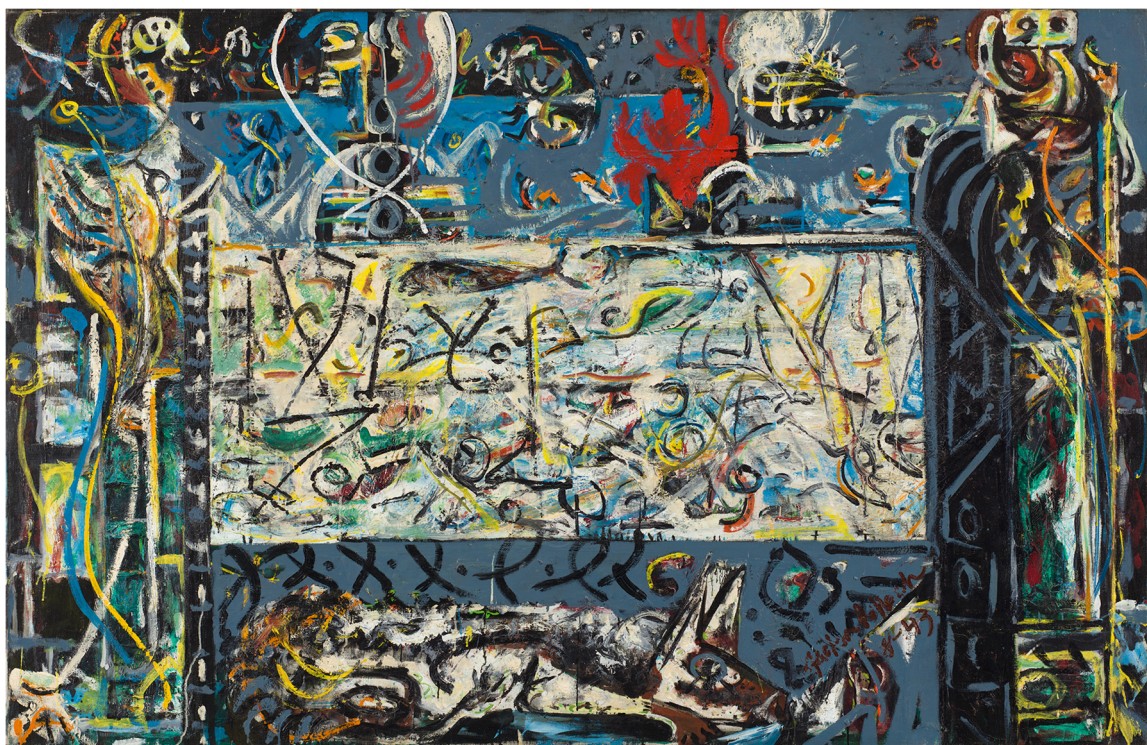

**Figure 6.** Jackson Pollock. *Guardians of the Secret*. 1943, oil on canvas, 48 3/4 in. × 75 in. San Francisco Museum of Modern Art. © 2023 The Pollock-Krasner Foundation/Artists Rights Society (ARS), New York. https://www.sfmoma.org/read/guardians-of-your-secret/. Accessed on 3 October 2022.

Finally, Barnett Newman was far more consistent with Pollock's loud championing of Native American influences, although he would take a more overtly intellectual course. Born in New York, Newman was not associated with the rugged Western lifestyles of Pollock and Still (Chipp 1968, p. 515). However, the mythic content of Native American beliefs and creative practice did agree with his more philosophical bent. Not only did Newman organize the exhibition "Northwest Coast Indian Painting" at Betty Parsons Gallery in 1946, he continually referenced Native American artwork as a precedent for modern abstraction in his contemplative articles and catalog essays (Landau 2005, p. 423). Newman opened his essay for "The Ideographic Picture"—an exhibition of abstract painting, which he also organized at Betty Parsons in 1947—with the following:

> The Kwakiutl artist painting on a hide did not concern himself with the inconsequential that made up the opulent social rivalries of the Northwest Coast Indian scene; nor did he, in the name of a higher purity, renounce the living world for the meaningless materialism of design. The abstract shape he used, his entire plastic language, was directed by a ritualistic will toward metaphysical understanding. The everyday realities he left to the toymakers; the pleasant play of nonobjective pattern, to the women basket weavers. To him a shape was a living thing, a vehicle for an abstract thought-complex, a carrier of the awesome feelings he felt before the terror of the unknowable. The abstract shape was, therefore, real rather than a formal 'abstraction' of a visual fact, with its overtone of an already-known nature. Nor was it a purist illusion with its overload of pseudoscientific truths (Landau 2005, p. 136).

Newman used the ceremonial and material culture of the Kwakiutl people to defend his own theory of the abstract sublime: "a ritualistic will toward metaphysical understanding" motivating the interplays of color and shape in their paintings on hide drums and other objects. Championing the Kwakiutl's spiritual purity was part of the artist's attempt to become "free from the weight of European culture," whose art he believed had become

too concerned with "the problem of beauty and where to find it" to create work that was truly sublime (Landau 2005, p. 139).[8] Newman's early works, such as the Metropolitan Museum's *Song of Orpheus* (1944–1945), draw from this formal and spiritual interest in Native arts and culture. These works are like cave paintings set loose in Petri dishes: free-floating glyphs and organic imagery, rife with symbolic potential.

Newman's Native-inspired biomorphism rhymes with Pollock's early works, as well as that of the other Abstract Expressionist "Myth-Makers" who were similarly interested in the connections between Native American spirituality and primal myths in the 1940s: Adolph Gottlieb, Robert Motherwell, and Richard Pousette-Dart, to name a few (Landau 2005, pp. 250–56, 422; Craven 1991, pp. 44–66). Reflecting on this early period, Lawrence Alloway writes,

> . . . in the '40s, mythology was seriously regarded as a key to the psycho-social order we share with world culture . . . .Thus, an artist with an interest in mythology could discover its enduring and fantastic patterns in his art and, at the same time, project his personal patterns out into the world (Landau 2005, p. 252).

Native American imagery connected the Abstract Expressionists to a tradition that seemed older, archaic, and mysterious to their modern sensibilities, answering the calls of Nietzsche and Jung to connect modern life to humanity's shared primal imagination (Landau 2005, pp. 423–25). While Still's experience with the Colville people did not lead to similar imagery, it is clear that some level of encounter with Native American creative practice was a foundational element of Abstract Expressionism, informing these three titans of the movement among many others.[9]

There are contesting social implications in the Abstract Expressionists' interest in Native American creative practices. It is significant that the Abstract Expressionists would speak of Native American influences on equal footing with—or in Newman's case, superior to—European ones. Art history professor David Craven writes, " . . . their decision to assimilate these visual languages on equal terms with European art during one of the most ethnocentric periods in U.S. history constituted a choice replete with ideological import, hence also with political implications." (Craven 1991, p. 60). Clyfford Still's work on the Colville Reservation can be seen as humanizing Indigenous people in a way unique for the time, his portraits genuine rather than exoticizing (Figure 7).[10] On the other hand, critic Billy Anania argues that Still's proposed art colony advocates more for white artists than for the rights of native peoples:

> Rather than aiding the tribe, the colony functioned as a purely educational experience for Washington State faculty and students, who observed the daily lives of their Indigenous subjects without embracing or materially supporting their lifestyle. In this context, the term 'art colony' carries particular implications (Anania 2020).

Neither did the other Abstract Expressionists seek political action on the behalf of Native people even as they borrowed from their cultures. While falling short of advocacy, the Abstract Expressionists asserted that a degree of participation in Native American traditions was something which made their work special, unique, superior. They maintained contact with Native art forms—studying them, attaining a scholarly mastery of them—and spun those influences to their own advantage.

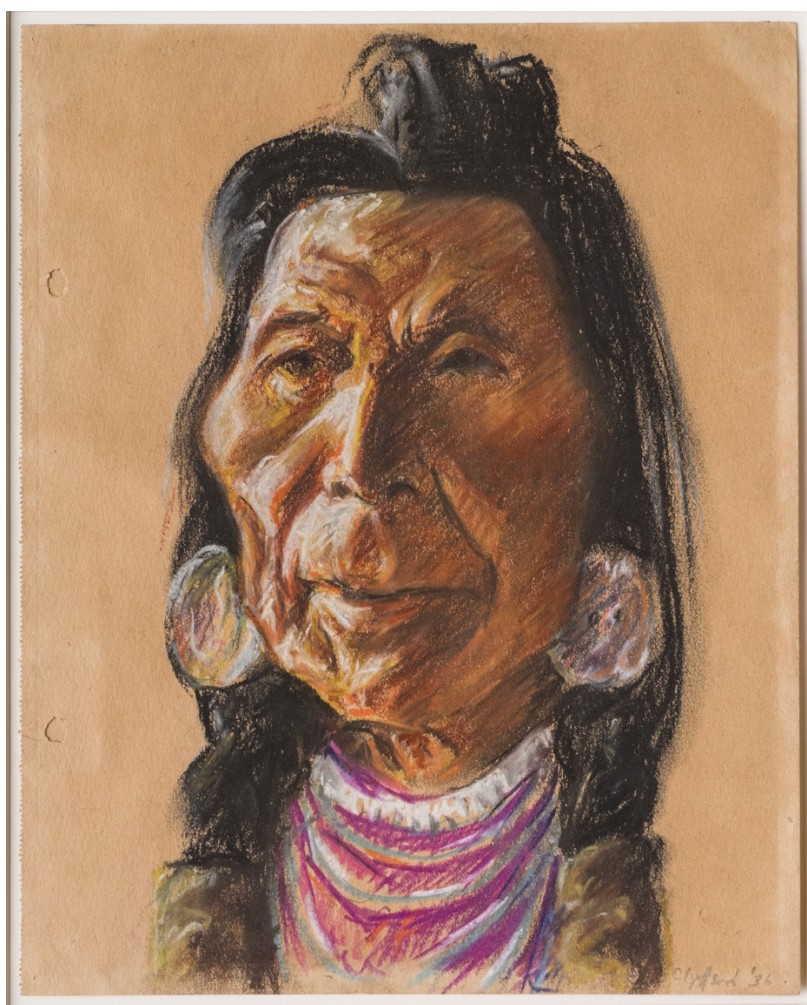

**Figure 7.** Clyfford Still. *PP-241/ Portrait of Willie Andrews*. 1936, pastel on paper, 12 1/4 × 9 3/4 in. The Clyfford Still Museum, Denver. © 2023 City & County of Denver, Courtesy Clyfford Still Museum/Artists Rights Society (ARS), New York. https://collection.clyffordstillmuseum.org/object/pp-241, accessed on 3 October 2022.

Whatever his racial leanings, the cowboy hero similarly walks the border between white "civilization" and Native "savagery." Usually the cowboy understands the mystic ways of local tribes, explaining them to his comrades. Sometimes he even attains a mastery of their language, a trust of certain tribal members. In *Broken Arrow*, a film broadly hailed as a signal in the changing white attitude towards Native Americans, Jimmy Stewart's character is the only white man willing to seek peace with the Apache people (Kilpatrick 1999, pp. 58–59). After learning their language and earning the trust of their leader Cochise, played by Jeff Chandler, Stewart works with other tribal leaders and white officials to secure a peace treaty (Figure 8). He even marries an Apache woman in the process.[11] In a less optimistic portrayal, Wayne's character in *The Searchers* is overtly racist to a degree that shocks even the other prejudiced characters. He continually calls his comrade Martin Pawley, played by Jeffrey Hunter, a "blanket head" for his 1/8 Cherokee blood, and insists that captives rescued from the Comanches "ain't white anymore." (Ford 1956). Up until the climax of the film, the viewer is not sure if Wayne will rescue his niece or kill her for being socially and sexually poisoned by her years spent with the Comanches, however unwillingly. In spite of his prejudice against Native cultures, Wayne's character is familiar with Comanche culture and language after years of living in the wilderness, giving him the knowledge and access he needs to hunt his niece so doggedly for the years-long span of the film. In each of these movies, the cowboy hero becomes a cultural bridge through his

contact and experience with Native American groups. Newman, Still, and Pollock, both in their stated influences and in the construction of their images by the art world, similarly became the cultural go-betweens for Native America and a predominately white art world, translating foreign rituals into a language intelligible to Euro-American Modernism.

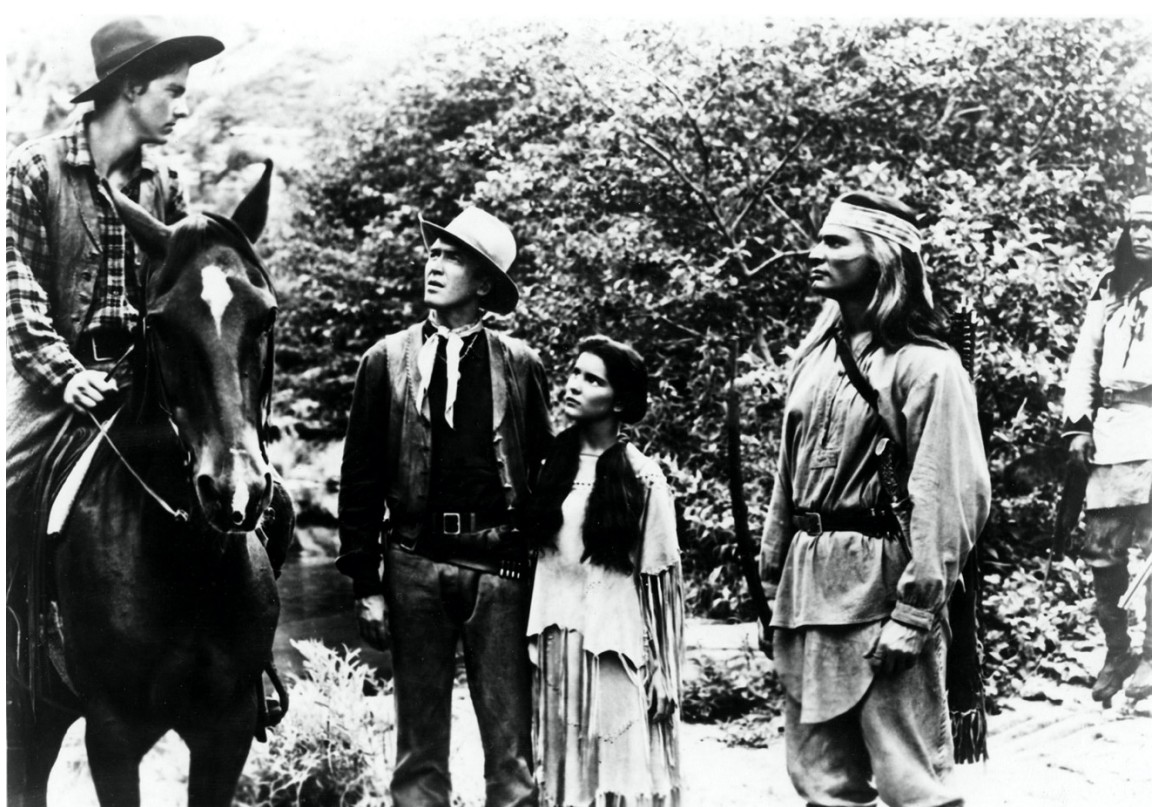

**Figure 8.** Jimmy Stewart (center) interceding between the Apaches, represented by Debra Paget and Jeff Chandler (right), and the whites in *Broken Arrow*, directed by Delmer Daves. 1950; Los Angeles, CA: Twentieth Century Fox, copyright 2023. Film © RGR Collection/Alamy Stock Photo.

This, too, coincides with Turner's frontier thesis. Turner writes that in order for white settlers to tame the Western wilderness, they had to first become "savage" as the Native Americans around them:

> The wilderness masters the colonist . . . . It puts him in the log cabin of the Cherokee and Iroquois and runs an Indian palisade around him. Before long he has gone to planting Indian corn and plowing with a sharp stick, he shouts the war cry and takes the scalp in orthodox Indian fashion (Turner and Faragher 1994, p. 33).

In this mantra of building civilization from the ground-up, the frontier narrative requires an appropriation and mastery of Native American practices in order to tame the wilderness of which they are supposedly a part.

Not only are these influences appropriated, they are overtaken from Native Americans by both the Abstract Expressionist and the cowboy hero. An underlying assumption for much of white America has been that Native American lifestyles would vanish in the face of "superior" white society. Turner continually equates Native American practices and lifestyles with "savagery," an earlier stage of "societal evolution" which is soon replaced by white civilization. This assumption of archaism has been reinforced by Western films, which in several ways depict Native Americans as relics of the past. While everything from their exoticized dress to their stilted speech patterns to their "barbaric" behavior stresses their anachronism in comparison to the "civilized" white heroes, the Native Americans' eventual defeat in most Westerns is a more insidious claim for their inferiority. Many

films feature an inevitable horse-chase in which a great number of Indians are shot off of horseback by a smaller number of white settlers. Implicit in the success of the whites over the Indians despite their numbers is their tactical superiority, the predictable retreat of the Indians signifying the inevitable disappearance of their culture. "Once white valor has been verified, the Indians simply disappear," writes Jacquelyn Kilpatrick in her book *Celluloid Indians: Native Americans and Film*. "The frontier having been crossed, the Indians vanish into the landscape, a part of the hostile world only the white hero can tame." (Kilpatrick 1999, p. 54).

Even in instances where some amount of sympathy is expressed for the plight of Native Americans, their demise is never questioned, only accepted. Art colonists and white patrons who supported Native artists at the turn of the twentieth century never stopped to ask if they could lobby for the protection of Indigenous sovereignty; rather, as Rushing states,

> The certainty that progress would eliminate the curious savages of the 'Painted Desert' made the collection (read preservation) of their material culture so urgent that even before 1900 the dominant culture's hunger for Pueblo and Navajo art could not be satisfied by the producers or their agents (Rushing 1995, p. 13).

Similarly, at the end of *Hondo*, the eventual demise of the Apache is put in no uncertain terms. As the company of settlers whom John Wayne has just saved from vengeful Apaches watches them vanish into the distance, Lt. McKay and Buffalo Baker remark that the US Army will soon come to bring the "end of the Apache." "Yep. End of a way of life, too," remarks Hondo. "Too bad. It's a good way. Wagons forward!" (Farrow 1953, 1:23:00–1:23:20). Even as Hondo is saddened by the doom of the Apaches—the people who raised him—he can only shrug and keep the white wagon train rolling. In the ebb and flow of white opinions on Native American culture, sympathy has often been born of a certainty that Indigenous ways are outmoded and, therefore, soon to disappear.

In this context, it is significant that Newman, Pollock, and the other "Myth-Makers" were interested in Native American sources as examples of *primal* Jungian myths: *archaic* stories, whatever relevance they had to modern life. It is also significant that even as Still painted the Grand Coulee Dam, watching it devastate the lifestyle of the Colville people, he did little beyond trying to bring other artists there to watch.

This is, admittedly, a harsh and reductive read on the Abstract Expressionists' interaction with Native American culture. The celebrity status of figures like Pollock allowed for an increased awareness of Native American life—still at the service of a white artist, but now brought before an international audience. There is a value in Still's sympathetic representation of a people in crisis, and for a young artist just beginning his career, perhaps that is all he was able to offer. As Craven points out, even as many of the Abstract Expressionists quietly espoused racial equality, their loudly stated apoliticism was necessitated by the reactionary policies of the McCarthy era, complicating any kind of statement or action that remotely smacked of the political left (Craven 1991, p. 61). By all accounts, the Abstract Expressionists had a genuine interest in the life and culture of Native American people. Even so, the ways in which their early work appropriated Native influences exists in the same tradition of cowboy Westerns whose heroes overtake Native culture to bring white society ever farther into Turner's frontier.

As with all points of crossover between these two movements, the question of influence cannot be determined with any certainty. However, given the previously cited prevalence of Western films during Abstract Expressionism's vogue, we can say that at least an unconscious belief in the fate of Native America was at play in the culture at large. Federal policy towards Indigenous Nations was certainly marked by this attitude as House Concurrent Resolution 108 was passed in 1953 to "terminate" all tribes "within the territorial limits of the United States" in an effort to "assimilate" Native peoples into American society. While this was seen as a kind of "affirmative action," the resolution dissolved not only Native sovereignty but many of the support systems put in place by tribal governments as well (Kilpatrick 1999, pp. 55–57). Turner's attitude, perpetuated in

visual culture, became a self-fulfilling prophecy: forcing the adaptation of Native peoples to white culture. This mindset runs so deep that the actions of painters, moviemakers, and politicians all point to each other in a cultural back and forth. The New American Painters' takeover of their Native American sources was likely shaped by these ideas as they were prominent in both cultural and political discourse.

At stake in all of these instances is the agency and perceived humanity of Native peoples. In Turner's day, the consequences were as plain as he wrote them: the so-called "Indian Question" answered by "a series of Indian Wars" which continuously and brutally shrank both Indigenous borders and populations. In film, the flattening of Indigenous cultures into the stoic, mystic, or war-like type created by Hollywood denies the diversity of individuals and cultural practices present in the many nations that make up what we have come to call Native America.[12] In art, even as the New York School asserted the aesthetic status of Native American artists through the inspiration of Pollock and others, too often the references to archaic practices, ancient objects, and primal Jungian myths keep Native American art in the past. The sand paintings and the totem poles had served their purposes, paving the way for white artists to take over. By claiming Native Americans as a part of their aesthetic lineage rather than players in the artistic present, the Abstract Expressionists and other American Modernists asserted that they were taking over for the new, modern era. Whatever the intentions of a painting like *Guardians of the Spirit* or a film like *Broken Arrow*, they still deny Native Americans the ability to live or create for themselves, granting this privilege instead to white artists, critics, actors, and directors. As the defeat of Native American nations is a defining tenant of the Frontier Thesis, both Abstract Expressionism and the Mythic West adhere to Turner's theory through their own kinds of conquest.

## 5. Conclusions

As the frontier narrative runs deep in Abstract Expressionism and Western films, it follows that both genres have become synonymous with America itself. Hollywood Westerns have proven a successful genre the world over, spinning off the entire European sub-genre of "Spaghetti Westerns" which numbers *The Good, The Bad, and the Ugly* (1966) among its contributions, a seminal film for the Western genre as a whole. As for the New American Painting, the US government itself acknowledged the movement's embodiment of American progress by sponsoring its international tours during the Cold War. As Eva Cockroft writes, the core value of individual expression became "ideal" for "present[ing] a strong propaganda image of the United States as a 'free' society as opposed to the regimented Communist bloc." (Cockroft 1985, pp. 128–29). It is notable, too, that Western films achieved their greatest popularity during this same period. In his book *Cowboys as Cold Warriors*, Stanley Corkin argues that

> Westerns of [the Cold War] period are concurrently nostalgic and forward looking. They look back upon the glory days of western settlement as they look ahead to the expression of U.S. centrality in the postwar world.
>
> ... [T]he historically resonant images found in Hollywood films provided a map for a great many Americans that helped them navigate the stresses and contradictions of Cold War life.". (Corkin 2004, pp. 9–10)

During an era of uncertainty, each of these genres became a way for Americans to reassure themselves of what it means to be American: a constant pursuit of freedom and innovation. Each become incorporated into American mythology because each elaborates so compellingly on the mythic American frontier.

This is the slippery nature of myth: the blend of fact and fiction perpetuating itself by influencing people's actions—in this case, historians, painters, and Hollywood directors—in turn creating *new* facts and *new* fictions. As many notable figures accepted the idea of American frontiering, part of United States history *did* come to be shaped by this mindset as it influenced imperial pursuits and technological progress. While Greenberg protested the extent to which his progressive theory of flatness was taken as gospel, it came to be one

of the movement's defining texts (Greenberg 1993, pp. 93–94). No matter how long it took Pollock to paint the Guggenheim *Mural*, Ed Harris' film has immortalized the myth of its completion in a single night. These myths came from a collection of facts and stories, from homesteaders and artists responding to challenges and from scholars attempting to make sense of them. Identifying the common frontier mindset between Abstract Expressionism and Western films should not limit these movements: each was shaped by a broad range of ideologies and circumstances beyond the Turner thesis. Rather, seeing the Turner thesis as a commonality helps us understand not only American painting and American film, but America itself: what it is, what it is not, and what we create it to be.

**Funding:** This research received no external funding.

**Data Availability Statement:** Not applicable.

**Conflicts of Interest:** The authors declare no conflict of interest.

## Notes

1.   The latter two examples, of course, referencing the foundational articles by Clement Greenberg and Harold Rosenberg, respectively (Greenberg 1993; Landau 2005, pp. 189–98).

2.   Other photographs of Pollock's early life included in the Smithsonian Institution's exhibition (The Smithsonian Institution 2012) attest to an authentic experience of the geographic and cultural American west during his childhood. However, another photograph of Pollock in the same garb and in the same location suggests that Pollock is not actually using the rifle in Figure 3, but posing for a series of cowboy portraits.

3.   Several such scholars are interviewed in (Jones and Rau 2021).

4.   The films I watched in preparing this article were those that appeared most often in scholarly sources as popular or noteworthy, particularly as discussed in (Desmaris and Smith 2017, pp. 36–64). Why and how the Apaches as led by Cochise and Geronimo became a common villain for some of the most successful Western films is a fascinating question that calls for further research.

5.   In order, this references the final shots of *The Searchers, Broken Arrow, Stagecoach, Hondo,* and *Fort Apache*, and I am sure many, many more.

6.   Pollock was likely also interested in the healing rituals of Navajo sand painting. Sand paintings are created for a sick or injured person: upon completion, the afflicted sits upon the painting to draw power from it before the painting is destroyed. Pollock's language of "being in the painting" likely derives from this physical contact of the person with the artwork. Rushing also states that drip painting may have also become a "shamanic process for healing" for Pollock as he fought with his own demons (Rushing 1995, p. 173).

7.   Images of an altar from the Zia Pueblo Snake Society were reproduced in the 1894 report of the *Bureau of American Ethnology*, a journal which Pollock read and had many issues (Rushing 1995, pp. 179–82).

8.   Newman particularly found the sublime in the mythic tragedies of Kwakiutl spiritual practice, particularly the *Hamatsa* dance in which one tribe member enacts the part of the cannibal spirit to be confronted and tamed by the tribe. This tempering of human cruelty likely appealed to Newman, the son of Jewish immigrants, in the days during and after the Holocaust. Rushing, "Impact of Nietzsche," in (Landau 2005, pp. 430–33).

9.   While Still often protested against any external sources shaping his work, Anfam proposes that the tragic content of his early paintings was likely bolstered by the same general interest in mythology—particularly tragic myth as a way of processing World War II—that inspired Newman to look to the *Hamatsa* cannibal dance. The network of literature circulating the scholarly and artistic circles of which Still was a part all point in this direction, Still likely absorbing the ideas however unwittingly or unwillingly. Anfam, "The Earth, the Damned," in (Landau 2005, p. 584).

10.   In a video introducing an exhibition of Clyfford Still's paintings from the Colville Reservation at the Clyfford Still Museum in 2016, art historian and Colville tribe member Michael Holloman states, "I don't see Clyfford Still relying upon the kind of expected generalizations or stereotypes. And the way that he wanted to present the native subjects, the people that were modeling for him, they're not reworked as other artists had done in the past to the audience's expectations of what Native people were supposed to look like. They were much more personal and intimate" Museum (Clyfford Still Museum 2016, 4:09–4:42).

11.   The casting of two white actors as the major Apache characters in this film—Chandler as Cochise and Debra Paget as Sonseeahray, Stewart's bride—is a very literal inhabiting of Native American culture by white Americans to render it digestible for a white audience.

12.   Kilpatrick gives a thorough overview of these stereotypes in "The Cowboy Talkies" chapter of her book, (Kilpatrick 1999, pp. 36–64).

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
