# Peer review of "Cowboys: Abstract Expressionism, Hollywood Westerns, and American Progress"

_arts_

Round 1
Reviewer 1 Report
Very interesting piece of research. Thought-provoking approach that finds parallels in the European context. I think that the article should be published as it brings forward an interesting approach on a well-researched topic.
Suggested edits:
page 21: first line: delete repeated word: This is, admittedly, is a harsh and reductive read
Author Response
Hello,
Thank you very much for your kind words on my article. Thank you also for identifying the error on page 21. I have fixed this error as well as other minor issues in syntax, diction, and grammar.
I appreciate you taking the time to review my work.
Reviewer 2 Report
I don’t find the comparison between Turner’s Frontier thesis and its supposed manifestation in Western films with Abstract Expressionist painting very convincing. The similarities that the author points out, such as the lonesome artist taming the blank canvas corresponding to the lonesome cowboy taming the wilderness or the cowboy hero and the painter appropriating Native American culture, seem strained and vague. The Turner thesis and its implications for the Western film genre also do not seem to me to be well captured. The point of most Westerns is less the celebration of taming the wilderness, but a dialogical negotiation of a supposed American identity which is located fluidly in between wilderness and civilization. Particularly the films the author mentions (Stagecoach, Fort Apache, Broken Arrow) are less about taming the West and more about the corruptions of White civilization from which the hero, partly, tries to dissociate himself. The point is that the frontier thesis is only a limited and reductive template for the films, which apply the frontier dynamics in diverse and complex ways often in relation to their contemporary socio-cultural context. A further point is that the frontier thesis and the Western genre constitute narratives, causally connected events and developments, which I don’t find particularly comparable to abstract expressionist painting, but this may just be my limited understanding of abstract expressionism (I am a film scholar, not an art scholar). In the attempt to “narrativize” abstract expressionism the author freely mixes references to the making of the paintings, their contents, the authorial approach to the films, and the contents of the films, which I find to make the argumentation more imprecise. The argument of the cowboy hero and the painter being a “go-between” between Native American and White Western culture is also not convincing. There are considerable differences between the role and status of the Western hero as a “man who knows Indians” within frontier mythology (see Richard Slotkin’s work) and the practice of Western appropriations of Native American art, and again this argument confuses a fictional construct with an actual artistic practice.
Author Response
Hello,
Thank you very much for your suggestions for improvement on my article.
- In response to your critique that the films I discuss are not solely manifestations of the Frontier Thesis, I have added emphasis that the Frontier Thesis was not the goal of these films, but rather a major influence whose effects are evident. I say the same for Abstract Expressionism: each of these genres exists beyond the Frontier Thesis, but recognizing its influence on these two cultural forms aids in our understanding of them as wholes. These additions can be found on page 13 and in the conclusion on page 24.
-
I have also expanded on the effect of the Turner thesis on the perceived archaism of Native Peoples in Western films by including additional analysis of Native American characters in film on page 21. This elaborates on Western film's connection to the Frontier Thesis in keeping with Jacquelyn Kilpatrick's scholarship in her book Celluloid Indians: Native Americans and Film (1999), among others.
- I appreciate your concern that I "freely mix references to the making of the paintings, their contents, the authorial approach to the films, and the contents of the film." In so doing, I was hoping to incorporate all viewpoints which created these cultural phenomena. Our understanding of both of these movements is informed as much by the paintings or films themselves as by the words of critics and scholars. While I state this in my introduction, I hope to have made this clearer throughout in editing for syntax and clarity.
- I also appreciate your concern that my paper "confuses a fictional construct with an actual artistic practice." I think this is the interesting part about this topic: while the Abstract Expressionists did not produce narrative paintings as did Western film directors, the public images that they and their critics cultivated aligns with the fictional characters created in Western films. I argue that this same character is prevalent in both because of Turner's idea of American progress. I have made this clearer in my conclusion on pages 23-4: our understanding of Abstract Expressionism, Western film, and America itself is informed just as much by fact as by fiction.
Thank you again for taking the time to review my paper and make suggestions. I feel these edits have made my paper stronger.
Reviewer 3 Report
The article is well suited for a special issue on “The Intersection of Abstract Expressionist and Mass Visual Culture,” and works fine as an introduction to the connections between the cultural imaginaries of the Western and Abstract Expressionism. Still, many topics raised in the piece could be explored in more depth.
Let me adduce a few examples/suggestions:
The notion of “the abstract sublime” is mentioned on p. 16 and would benefit from a consideration of Robert Rosenblum’s 1961 article “The Abstract Sublime,” and subsequent book Modern Painting and the Northern Romantic Tradition: Friedrich to Rothko (1975), especially in light of the rückenfigur motif (i.e. Friedrich) which is mentioned on p. 13. Overall, Rothko’s colorfield compositions, evocative of Western sunsets, warrants commentary.
A reflection on how ‘western-ness’ functions as a unifying trope for stylistic diversity (the frenzied patterns of Pollock’s action paintings and the meditative, contemplative effect sought by colorfield painters such as Rothko and Still).
A reflection on Clement Greenberg’s distinction between Avant-garde and Kitsch, vis-à-vis Abstract Expressionism and Hollywood Westerns.
How does Jackson Pollock’s statement “I am nature” fit with the author’s thesis that Abstract Expressionism, in common with the Hollywood Western, was dedicated to the taming of the West?
The correlations between Abstract Expressionism and a very small selection of Westerns come across as a bit redundant. One way to strengthen these comparisons could for example be to elaborate further on the issue of scale (i.e. the introduction of various widescreen formats coeval with the grand canvases of AE).
Author Response
Hello,
Thank you very much for your kind words and suggestions for improving my article. I have made the following revisions:
- On page 11, I have incorporated information from John Belton's book Widescreen Cinema (1992) in response to the need for more discussion on the use of widescreen in Western films. Not only does the "grand scale" of widescreen film correlate to the larger sizes of Abstract Expressionist canvases, it also connects to Robert Rosenblum's ideas of the sublime as confronting "boundlessness" in Abstract Expressionism, as suggested.
- In keeping with the above, the discussion of the "cowboy rückenfigur" on page 13 is more directly connected to the Romantic era, and therefore to the feeling of awe before nature. This strengthens the connection between Western films and Turner's emphasis on the wilderness.
- A few sentences have been added on page 9 reflecting on how the Western ideal unifies Still's and Pollock's vastly different paintings through both scale and a "plain-speaking attitude" of paint as paint.
Thank you again for taking the time to review and recommend my article. I feel that these edits have made for a stronger paper.
Round 2
Reviewer 2 Report
My general problems with the article's understanding of the Western's use of the frontier thesis remain, even if some arguments are attenuated accordingly. I don't quite see the connections that the article makes between the films and the paintings or their discursive frames, but this may be due to my lack of expertise in arts scholarship.